# The Mysteries around the BCL-2 Family Member BOK

**DOI:** 10.3390/biom10121638

**Published:** 2020-12-04

**Authors:** Raed Shalaby, Hector Flores-Romero, Ana J. García-Sáez

**Affiliations:** Institute for Genetics and Cologne Excellence Cluster on Cellular Stress Responses in Aging-Associated Diseases (CECAD), University of Cologne, Joseph-Stelzmann-Straße 26, 50931 Cologne, Germany; raed.shalaby@uni-koeln.de (R.S.); hector_uniupv@hotmail.com (H.F.-R.)

**Keywords:** BOK, MOMP, BCL-2 family, apoptosis

## Abstract

BOK is an evolutionarily conserved BCL-2 family member that resembles the apoptotic effectors BAK and BAX in sequence and structure. Based on these similarities, BOK has traditionally been classified as a BAX-like pro-apoptotic protein. However, the mechanism of action and cellular functions of BOK remains controversial. While some studies propose that BOK could replace BAK and BAX to elicit apoptosis, others attribute to this protein an indirect way of apoptosis regulation. Adding to the debate, BOK has been associated with a plethora of non-apoptotic functions that makes this protein unpredictable when dictating cell fate. Here, we compile the current knowledge and open questions about this paradoxical protein with a special focus on its structural features as the key aspect to understand BOK biological functions.

## 1. Introduction

The proteins of the BCL-2 family are the main regulators of the intrinsic apoptotic pathway and play a pivotal role in tumorigenic cell removal and cancer treatment effectiveness [1,2,3]. They form a complex interaction network that controls the key step of mitochondrial outer membrane permeabilization (MOMP), which is considered the point of no return in the cell death decision. MOMP enables the release of several apoptotic factors, such as cytochrome c and SMAC, from the intermembrane space to the cytosol to induce apoptosome formation, caspase activation, and apoptotic cell death execution [4]. 

There are around 20 members in the BCL-2 family, which are typically classified into three groups according to their impact on cell death and the presence of up to four conserved BCL-2 homology (BH) motifs: (i) the pro-apoptotic effectors, including BAK and BAX, and to which BOK has been traditionally assigned, contain BH1-BH4 motifs and directly elicit MOMP, (ii) the anti-apoptotic members (like BCL-2, BCL-XL, and MCL-1), which also possess all four BH motifs and primarily function by inhibiting MOMP; (iii) the BH3-only proteins (like BID, BIM, or BAD), which promote apoptosis either by directly activating the pro-apoptotic effectors and/or by sensitizing to MOMP by blocking the anti-apoptotic members [5,6]. 

Despite their functional divergence, both pro-apoptotic and anti-apoptotic multidomain members fold into a similar globular structure: two central hydrophobic α-helices surrounded by six or seven amphipathic α-helices [7,8]. This characteristic fold gives rise to a hydrophobic groove defined by the helices α2–α5. Generally, the proteins of the BCL-2 family bind to each other via BH3-into-groove dimeric interactions, in which the BH3 motif of one protomer binds to the hydrophobic groove of another protomer. This gives rise to a mixture of BCL-2 homo- and heterodimers, whose balance regulates the apoptotic outcome [9,10]. Under this premise, several models have been proposed that differ from each other in the binding affinities attributed to different family members (Figure 1). The direct model or MODE 1 of inhibition proposes that anti-apoptotic proteins repress apoptosis by preferentially neutralizing the BH3-only activators, which are required for the activation of pro-apoptotic effectors [10,11,12,13]. In contrast, the indirect model or MODE-2 postulates that the inhibitory effect of the prosurvival BCL-2 is mainly driven by blocking the pro-apoptotic effectors, which would be constitutively active [14,15]. In the last scenario, the main role of the BH3-only proteins would be to release constitutively-active pro-apoptotic effectors from the anti-apoptotic proteins, hence “sensitizing” the cell to undergo apoptosis, as proposed in the unified model that merges MODES 1 and 2 [16]. Additionally, anti-apoptotic proteins can inhibit apoptosis by keeping pro-apoptotic effector proteins inactive through continuous retrotranslocation from the mitochondrial surface into the cytosol [17], MODE 0. The embedded-together model was the first one to consider the role of the membrane environment by proposing that the insertion into the mitochondrial outer membrane (MOM) triggers a conformational change that alters the interaction surfaces between BCL2 family members [18]. The quantification of the binding affinities within the membrane environment then led to the integrated model, which defines that the anti-apoptotic members bind preferably to the activators instead of the apoptotic effectors, in agreement with the bi-modal mechanism proposed in the hierarchical model [10,12]. Both these later models incorporate the ability of BAX-like effectors to auto-activate (Figure 1A).

Besides BH3-into-groove interactions, a plethora of non-canonical surfaces (e.g., rear binding site, N-terminal helix α1 and tail anchoring domain) [20,21,22,23] and mechanisms (Figure 1A), membrane lipid composition [10,24,25], posttranslational modifications such as phosphorylation, proteolytic cleavage, ubiquitination, and proteasomal degradation [26,27] have been proposed to modulate the BCL-2 interaction network and thus the cell fate. Independently of the model, induction of mitochondrial apoptosis leads to the activation of BAX-type proteins and apoptotic pore formation. The structural changes driving BAX-type proteins from an inactive conformation to a fully activated structure are usually divided into (I) Early activation steps (including TM dislodgement and N terminal exposure). (II) BH3 exposure occurs due to BAX reorganization in two functionally different parts, named “core” and “piercing” domains. (III) Oligomerization and (IV) Pore formation. Currently, it is mostly accepted that BAX and BAK mediate MOMP with the formation of heterogeneous toroidal pores of tunable size (Figure 1B). We speculate that BOK may also follow a similar mechanism of action. This scenario becomes even more complex in light of the additional non-apoptotic roles that have been proposed for the BCL-2 protein, including mitochondrial morphology, calcium homeostasis, unfolded protein response, DNA damage response, whole-cell metabolism, and autophagy [28].

## 2. BOK 

BCL2-related ovarian killer (BOK) is a highly conserved BCL-2 family member that preserves both sequence and structure homology to the multi-domain BCL-2 family members (Figure 2) [29]. BOK was first identified using a yeast 2-hybrid screen of a rat ovarian fusion cDNA library, with the anti-apoptotic MCL-1 as the bait [30]. In this study, BOK was first clustered within the effector subgroup of the BCL-2 family proteins, due to key similarities with the apoptotic effectors BAK and BAX. Primarily, it was shown that BOK contains several BH motifs, induces apoptosis upon overexpression in cells, and interacts strongly with the anti-apoptotic proteins MCL-1 and BFL-1, but not BCL-XL and BCL-2 [31].

However, despite the seemingly pro-apoptotic nature of BOK, new insights about its proteasomal regulation and endoplasmic reticulum (ER) targeting challenged its classification. In 2012, Lee et al. showed that BOK is ubiquitously expressed in various tissues but the expression level is higher in reproductive tissues such as the ovary, testis, and uterus [34,35]. In that study, they also produced the first BOK^-/-^ mice that appeared normal and fertile and displayed normal tissue architecture. In contrast to other BCL-2 proteins, BOK was found to localize more to the membranes of the ER and the Golgi apparatus than to mitochondria [36]. Interestingly, the C-terminal transmembrane domain (TMD) of BOK was shown to be necessary and sufficient for this targeting. As shown in Figure 2, in contrast to the TMD of BAX, BAK, and BCL-X, the TMD of BOK contains two positively charged residues: Arg199 and Lys200 in the middle of the α9 helix that reduces its hydrophobicity. However, it lacks the two successive electropositive residues just after the α9 helix that exist in BAX (KK), BAK (RR), and BCL-XL (RK). This might decrease the degree of mitochondrial localization of BOK, as it was reported that a net positive charge of ≥+2 at the C-terminal end is required for mitochondrial targeting [36,37,38].

On the transcription level, BOK was shown to be cell cycle-regulated via the binding of the transcription factor E2F1 to a conserved E2F-binding site in the BOK promoter region [39]. Furthermore, the promoter region of the BOK gene contains a hypoxia response element that binds to hypoxia-inducible factor proteins leading to an induction of BOK expression upon placental hypoxia or oxidative stress [40]. It was also reported that BOK expression is post-transcriptionally downregulated by a mechanism that involves conserved (AU/U)-rich elements in its 3′ Untranslated Region (UTR) [41]. TRIM28 (Tripartite motif-containing 28) was then shown to associate with U-rich elements in BOK 3′UTR to reduce its expression level through mRNA destabilization. 

Currently, the prevalent view accepts that BOK is a pro-apoptotic BCL-2 member as it was observed in different cell systems that BOK overexpression induces MOMP, caspase-3 activation, and apoptosis [30,36,39,42,43]. However, whether BOK elicits its pro-apoptotic function as a BAX-like effector remains debated, as we discuss below. Moreover, BOK appears to regulate multiple non-apoptotic processes including mitochondrial dynamics, calcium signaling, and metabolism via poorly understood mechanisms [44,45].

### 2.1. Is BOK a Canonical Pro-Apoptotic Effector like BAK and BAX?

The pro-apoptotic multidomain BAX and BAK have been traditionally considered as the unique MOMP effectors of the intrinsic apoptotic pathway. This notion was founded on the fact that BAK^−/−^/BAX^−/−^ cells are resistant to most pro-apoptotic stimuli [46]. Despite this, BOK was classified, although often overlooked, as a pro-apoptotic effector due to the high amino acid sequence similarity to both BAK and BAX, and because its overexpression was able to trigger apoptosis in various cell lines [30,31,36,47,48].

The fact that a fraction of BAK^−/−^BAX^−/−^ mice can survive to adulthood with normal morphogenesis of multiple tissues suggested that either apoptosis is executed by another protein or a different cell death pathway is compensating for apoptosis [49]. The functional redundancy between BOK and BAK/BAX was confirmed when it was shown that BOK^−/−^BAK^−/−^BAX^−/−^ mice had more severe defects and died earlier than BAK^−/−^BAX^−/−^ mice which may indicate that BOK has overlapping roles with BAK and BAX in developmental cell death [50]. Importantly, it was shown that developmental apoptosis loss in those mice was not substituted by other forms of cell death.

In contrast to the mostly cytosolic BAX and mitochondrial BAK, BOK is mainly targeted to the ER under healthy conditions. Based on this, it was proposed that BOK could elicit its pro-apoptotic function through a different mechanism to that of BAK and BAX. A number of studies reported that the pro-apoptotic activity of BOK is mainly regulated by the proteasome/ER system and occurs unresponsive to antagonistic effects of the anti-apoptotic BCL-2 proteins [42,51]. In general, these reports support a functional relevance of the subcellular localization of BOK at the ER, in agreement with the observation that ER stress (the saturation of ER capacity to fold proteins) can promote apoptosis [36,52,53]. In 2015, Carpio et al. showed that BOK^−/−^ cells had reduced susceptibility to undergo intrinsic apoptotic pathway in response to various ER stress stimuli with no observed differences in response to typical apoptotic stimuli (etoposide, staurosporine, or ultraviolet irradiation) [51]. They also reported that BOK expression can rescue the impaired response of BOK^−/−^ cells but not BAK^−/−^BAX^−/−^ cells and hence suggested that BOK requires downstream BAK or BAX activation to induce mitochondrial apoptosis, a hypothesis that has been challenged by other studies [42].

In 2016, Llambi et al. studied BOK function in Mouse Embryonic Fibroblasts (MEFs) upon doxycycline-induced expression [42]. They reported that BOK was not detected in wildtype MEFs even after induction of expression unless cells were treated with proteasome inhibitors. Silencing of endogenous BOK increased cell survival in response to proteasome inhibition, while it did not affect other canonical apoptotic stimuli. The authors identified multiple lysine residues in BOK that served as ubiquitination targets, as well as BOK-interacting proteins involved in the regulation of its pro-apoptotic activity. Based on these findings, they suggested that, in healthy cells, endogenous BOK is bound to gp78, which targets it for proteasomal degradation resulting in almost undetected cellular levels. ER stress then leads to the saturation of the Endoplasmic-reticulum-associated protein degradation (ERAD) system and subsequent BOK stabilization and translocation to the MOM to induce apoptosis. In light of this model, BOK could be an important player linking ER stress to intrinsic apoptosis.

In model membranes, BOK exhibits membrane permeabilizing activity similar to that of BAK and BAX. Recently, Fernández-Marrero et al. reported that C-terminally truncated recombinant BOK (BOK∆C) was able to permeabilize liposomes with a composition that mimics the mitochondrial outer membrane, but not the ER [54]. The main difference between those model membrane compositions is the presence of cardiolipin in the MOM-like samples. Cardiolipin is a negatively charged phospholipid that induces a negative monolayer curvature, and has been previously proposed to play a role in BAX pore formation [55]. Interestingly, the pores formed by BOK∆C were stable over time and large enough to allow the flux of large molecules like cytochrome c (12 kDa) and allophycocyanin (104 kDa). The membrane permeabilizing activity of BOK∆C was strongly affected by the lipid intrinsic monolayer curvature, suggesting the formation of toroidal pores with the participation of lipids at the pore edge, like those of BAX and BAK [56,57,58]. BOK∆C activity on liposomes was augmented by cBID and was not inhibited by BCL-XL. Still, BOK∆C was not able to release cytochrome c from mitochondria isolated from BAK^−/−^/BAX^−/−^ cells, not even with the presence of cBID or heat. This might be attributed to the presence of anti-apoptotic members on the mitochondrial membrane that inhibits BOK activity. Another possible reason could the lack of the C-terminal tail-anchor, which may be specially required for the activity of BOK on isolated mitochondria.

The interaction of BOK with the different BCL-2 family members has also been a matter of debate. First, it was reported that overexpressed BOK interacts with MCL-1 and BFL-1 but not BCL-2 and BCL-XL [30]. Later, Echeverry et al. showed that ectopically expressed BOK co-immunoprecipitated only with itself but not with any other of the tested BCL-2 family members (BCL-2, BCL-XL, MCL-1, BAK, BAX) and that this interaction was dependent on key residues in the BH3 domain [36]. Moreover, while BAK and BAX constitutively retro-translocate from mitochondrial membranes, this activity has not been observed for BOK [17]. This might not be surprising considering the preferential targeting of BOK for the ER, its constitutive degradation, and its apoptotic activity seemingly independent of other BCL-2 proteins [42,51,59]. In summary, although the interaction preferences of BOK with other BCL-2 family members have not been fully defined yet, the body of evidence so far supports a distinct pattern of binding partners compared to BAX and BAK.

### 2.2. Insights from BOK Structure

The low solubility and stability of BOK have hindered its structure determination for a long time compared to other BCL-2 family members [19,60]. A recent x-ray crystallography study used a version of chicken BOK, which shares 81% sequence identity with the human orthologue, lacking both the C-terminal membrane anchor helix and the first 18 residues of the N terminus (ΔN18ΔC32) to enhance the solubility of the protein and minimize structural disorder that hinders crystallization [50]. The resulting crystal structure revealed that soluble BOK adopts the typical BCL-2 fold similar to BAK, BAX, and the anti-apoptotic BCL-2 family members, which consists of two central hydrophobic α-helices surrounded by six amphipathic α-helices (Figure 3). The asymmetric unit contained two molecules of BOK that were generally similar except in the conformation of the loop between helices α2-α3 and the structure of helix α3, which was defined only in one of them. This results in two different architectures for the hydrophobic groove, one accessible and one occluded by residues Q92 and Q113. In addition, the residues in the groove region had higher B-factor values than the rest of the protein residues suggesting larger flexibility and dynamics [61]. Importantly, the internal cavities around helix α2 of BOK were similar to those found in the crystal structures of BAK and BAX when bound to BH3 peptides, this is, the activated forms [62,63,64]. As the exposure of the BH3 domain in the α2 helix of BAK/BAX is required to initiate oligomerization and subsequent membrane permeabilization, these findings may explain, at least partially, the constitutive activity of BOK.

The soluble structure of human BOK was also recently determined by nuclear magnetic resonance (NMR), which offers the advantage of providing information about protein dynamics and intermolecular interactions [65]. The additional structural features of BOK revealed by this study further explain its auto-activation and the distinct pattern of binding to other BCL-2 family members. First, BOK presents an atypical hydrophobic groove architecture that hinders binding to BH3 domains. In contrast to other BCL-2 proteins, the hydrophobic groove of BOK appeared to be collapsed and mostly made up of loop structures. This region is likely to undergo conformational exchange, as indicated by the dynamics detected by NMR, and in agreement with the two distinct groove conformations resolved by x-ray crystallography of chicken BOK [50]. In addition, the occlusion of the groove by helix α3 limits the access to the conserved small hydrophobic pockets P0 and P1, which are critical for the BH3-into-groove interaction. The lysine at position 122 (K122) of BOK also has a less positive charge than the conserved arginine in all other BCL-2 family members and therefore makes a less stable salt bridge for binding with the conserved Aspartate of the BH3 ligand. All of these may explain the weak affinity between BOK and a BH3 peptide of BID, which was 30- to 300-fold lower than that of BAK [66].

Second, BOK has a glycine residue (G35) in the middle of the α1 helix which was proposed to act as a helix breaker that reduces the stability of the BCL-2 fold. This would result in a higher tendency to spontaneously undergo the conformational change associated with activation of the membrane permeabilizing activity of BAX-like effector proteins. In agreement with this hypothesis, a mutant version of BOK, G35A, exhibited a significantly higher melting temperature and induced less permeabilization of liposomes than the wild type protein. BOK G35A was not able either to induce apoptosis in BAK^−/−^/BAX^−/−^ cells in contrast to wild-type BOK. These observations suggest that the destabilizing effect provided by G35 may favor spontaneous membrane binding and subsequent permeabilization. 

Altogether, the structural evidence so far supports a role for BOK as a BAX-like MOMP effector, where its membrane permeabilizing activity is driven by the intrinsic conformational instability of the protein. It should be noted that the structural data available are only relevant for the structure of BOK in solution, which is presumably distant from its active, membrane-bound form. The structure of BOK in the membrane environment remains unknown, as for the other BCL-2 family proteins, and is a key question in the field. In this regard, a recent study has revealed a role for the C-terminal membrane anchor of BOK in the interactions with itself and with other BCL-2 family members, which together with other non-canonical surfaces proposed for BAX and BAK, could define the alternative BOK interaction network [20,21,22,23,67].

### 2.3. Non-Apoptotic Functions of BOK

In addition to its role in mitochondrial apoptosis, BOK, like many other BCL2 proteins, has been associated with the regulation of additional cellular processes including mitochondrial dynamics, morphology, and bioenergetics. This extends the role of BOK as an important piece in the puzzle of apoptosis and highlights the need to better understand the molecular mechanisms of BOK to obtain an integrative view of its biological function in health and disease.

#### 2.3.1. Role of BOK in Calcium Signaling

In contrast to other Bcl-2 family members which are located at the mitochondria, BOK is believed to predominantly reside at the ER and the Golgi [35]. There, BOK binds with high affinity to inositol 1,4,5-trisphosphate receptors (IP3Rs), especially IP3R1 and IP3R2, which are involved in intracellular calcium signaling. In agreement with this, caspase 3-mediated cleavage of IP3R1 was more readily detected in BOK^−/−^ cells than in wild-type cells upon exposure to staurosporine [68]. The interaction between BOK and IP3R was suggested to be mediated by the BH4 domain of BOK, as mutating to alanine residues in this motif abolished binding. This interaction protects BOK from degradation and mature BOK could not be displaced from IP3Rs with agents that stimulate apoptosis or ER stress [69]. Despite binding to IP3Rs, BOK does not seem to affect IP3R-mediated calcium mobilization activity or calcium influx into mitochondria [44,69]. In contrast to this, BOK^−/−^ primary cortical neurons exhibit a decreased, but prolonged rise in cytoplasmic calcium levels in response to N-Methyl-D-aspartate (NMDA) excitotoxicity [28,70]. In light of these contradictory observations, the contribution of BOK to the regulation of IP3R calcium dynamics deserves further investigation.

#### 2.3.2. BOK Contribution to Mitochondrial Morphology

The proteins of the BCL-2 family are well known for their role in regulating mitochondrial dynamics, morphology, and function [28]. Mitochondrial morphology is mainly controlled by the balance between both mitochondrial fusion and fission. A recent study by Schulamn et al. has reported that deletion of BOK causes a cellular phenotype with fragmented mitochondria [44], which was attributed to a decrease in the mitochondrial fusion rate. Stable expression of either wild type or a version of BOK deficient in IP3R binding restored the normal mitochondrial morphology, indicating that the mitochondrial fragmentation was specifically due to the loss of BOK. Of note, mitochondrial fragmentation has also been observed with the deletion of other BCL-2 family proteins, and related to interactions with the mitochondrial fission or fusion machinery (e.g., BAK/BAX with Mitofusin 2) [71,72]. Although there is no indication of interaction between BOK and any mediator of mitochondrial dynamics, it cannot be discarded that the effect of BOK on mitochondrial fusion is indirect, through interaction with other BCL-2 proteins, like MCL-1, which interacts with both BOK and Drp-1 [44,73]. In agreement with this, a recent study suggests that the C-terminal domain of MCL-1 can target BOK to the mitochondria and increase the mitochondria/ER contact sites [67].

#### 2.3.3. BOK Regulates Uridine Metabolism

A recent study by Srivastava et al. has revealed that BOK positively regulates uridine monophosphate synthetase (UMPS), an enzyme involved in uridine biosynthesis [45]. Binding between two proteins was discovered using a yeast 2-hybrid screen using mouse BOK as bait and was further confirmed using immunoprecipitation in MEFs. The interaction is specific to BOK and mediated via the BH3 domain as site-directed mutagenesis of key residues in the BH3 domain abolished it and a BIM mutant bearing the BH3 domain of BOK was able to interact with UMPS, while BIM itself could not. UMPS is the key enzyme involved in the conversion of the chemotherapeutic drug 5-fluorouracil (5-FU) to its toxic metabolites in cancer cells [74]. In line with this, BOK^−/−^ MEFs were shown to be more resistant to 5-FU compared to wild-type cells and down-regulation of BOK was detected in cell lines already resistant to 5-FU. These findings open possibilities for using BOK as a biomarker for 5-FU resistance or developing BOK mimetics for sensitizing 5-FU-resistant cancers. 

### 2.4. BOK as a Prognostic Marker

The tumor suppressor function of BOK was first proposed based on the finding that the genomic locus containing the BOK gene was frequently deleted in different human cancers [75]. Unexpectedly, among the pro-apoptotic family members, BOK is one of the most frequently deleted genes in cancer cells [76]. It was also shown that BOK is downregulated in patients with late-stage compared to early-stage non-small cell lung carcinoma (NSCLC) and high BOK levels could predict extended patient survival [41]. A recent study has shed new light on the role of BOK in colorectal cancer (CRC) progression and its correlation with clinical outcome [77]. Stage II and III CRC patients had significantly reduced the expression level of BOK in their tumors compared to normal tissues, which opens the possibility of using BOK as a prognostic marker in CRC. However, the observation that BOK mRNA expression was not prognostic in CRC might imply that BOK is mainly post-translationally regulated. Moreover, a high level of BOK expression was correlated with reduced survival and disease recurrence, which is not in agreement with previous findings in other cancers [75]. This may be attributed to the existence of different roles of BOK in tumor establishment and recurrence, probably because of its involvements in different cell regulation processes apart from apoptosis. For instance, high expression of IP3 receptors was shown to be associated with metastasis formation and aggressiveness of different tumors and can be then used as a biomarker [78]. In addition, BOK was shown to localize to the nuclei of proliferating trophoblast cells in early placental development, where it regulates cyclin E1 expression [77,79]. Accordingly, increased BOK expression in these tissues was associated with trophoblast hyperproliferation and the development of preeclampsia [80]. Interestingly, a splice variant of BOK that lacks part of the 5′ UTR, the BH4, and part of the BH3 domain was reported in preeclampsia patients [76,81]. Despite these promising findings, further work will be necessary for the future to establish BOK as a prognosis marker in cancer.

## 3. Summary

After years of research efforts, it has become clear that the molecular features of BOK are different from conventional BAX-type effectors, yet its role in the cell remains unsettled [82]. BOK has been reported to promote MOMP and apoptosis in response to endoplasmic reticulum stress via a proteasome-dependent regulatory mechanism that differs from the classical apoptotic effectors BAK and BAX [42,51]. However, our knowledge of the mode of action of BOK is far more limited than that of BAK and BAX. While there is literature describing the topology of active BAK/BAX in the membrane, we still fail to understand how BOK interacts with cellular membranes. The similar ability of BOK and BAK/BAX to form membrane pores [54,83,84] raises the question of whether these proteins cooperate to form mixed assemblies at the MOM and whether these assemblies include mixtures of homodimers or heterodimers of BAK/BOK and BAX/BOK, as previously suggested for BAK and BAX [85,86]. Further research is needed to understand the structural and functional consequences of potential cooperation between BAK, BAX, and BOK, and whether BOK induces large pore assemblies compatible with mitochondrial DNA release, as reported for BAX and BAK [87,88]. Furthermore, BOK overexpression promotes morphological changes in mitochondria [44], ER, and Golgi [89], raising the possibility that BOK could have a role in shaping the membranes of these organelles. This could be especially the case at the contact sites between mitochondria and ER [67,90]. Considering the distinct role of BOK within the BCL-2 interactome and its numerous non-death functions, BOK remains an enigmatic protein at the cross-roads between calcium signaling, mitochondrial morphology, metabolism, and cell death.

## Figures and Tables

**Figure 1 biomolecules-10-01638-f001:**
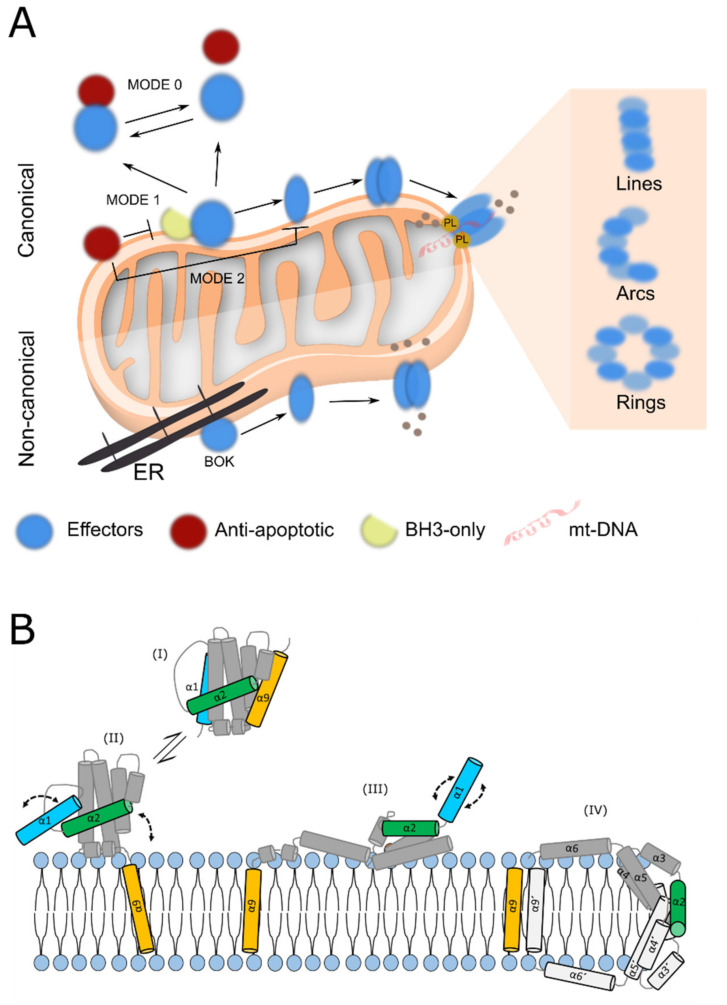
The BCL-2 family of proteins. (**A**) Regulation of apoptosis by the BCL-2 interaction network. Top, canonical BAX/BAK activation. Activation of BAX-type proteins at the mitochondrial outer membrane (MOM) by the BH3 only proteins induces their oligomerization, formation of supramolecular structures (lines, arcs, and rings), and pore formation with the consequent release of apoptogenic factors. The apoptotic repressors, block this process by either interacting with BH3 only proteins (MODE1) or with BAX-type proteins in the membrane (MODE 2) or translocating them to the cytosol (MODE 0). Bottom, non-canonical cell death mechanism elicited by BOK. Under cellular stress, BOK can avoid its proteasomal degradation, directly eliciting permeabilization of the MOM. PL: phospholipids; grey balls: apoptogenic factors; mtDNA mitochondrial DNA. (**B**) Model for the putative structural reorganization of BOK during the activation process based on our current knowledge of BAX. (I) Protein disposition in solution. BAX is represented with nine cylinders corresponding to its nine α-helixes and based on [19]. (II) BAX early activation steps, including TM dislodgement and N terminal exposure (depicted in orange and cyan respectively). (III) BAX reorganization into two different parts (dimerization and piercing domains) and BH3 domain exposure (depicted in green). (IV) Oligomerization and pore formation, structural representation of membrane-embedded BAX in the context of the toroidal pore (clamp model, based on [10]. One monomer is shown in grey (α1-9) and the other is depicted in dark grey (α1’-9´). The relative orientation of the helices 9 remains unresolved.

**Figure 2 biomolecules-10-01638-f002:**
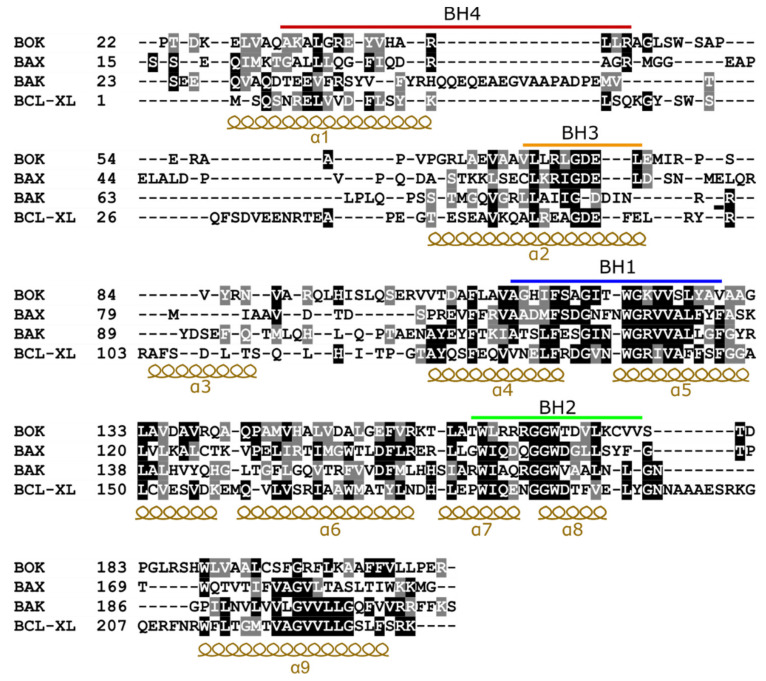
Structure-based sequence alignment of a subset of Bcl-2 family members. For aligning the sequences, the structure of BOK (PDB: 6CKV), BAX (PDB: 2K7W), BAK (PDB: 5FMI), and BCL-XL (PDB: 1BXL) were superimposed. The alignment was performed using UCSF Chimera software [32]. Identical residues are highlighted in black, while similar ones are highlighted in gray. The BCL-2 homology (BH) motifs are marked with bars and sequence regions corresponding to α-helices are indicated. The sequences of the transmembrane helix (α9) were aligned using Clustal Omega [33] as this region was not resolved in the structures.

**Figure 3 biomolecules-10-01638-f003:**
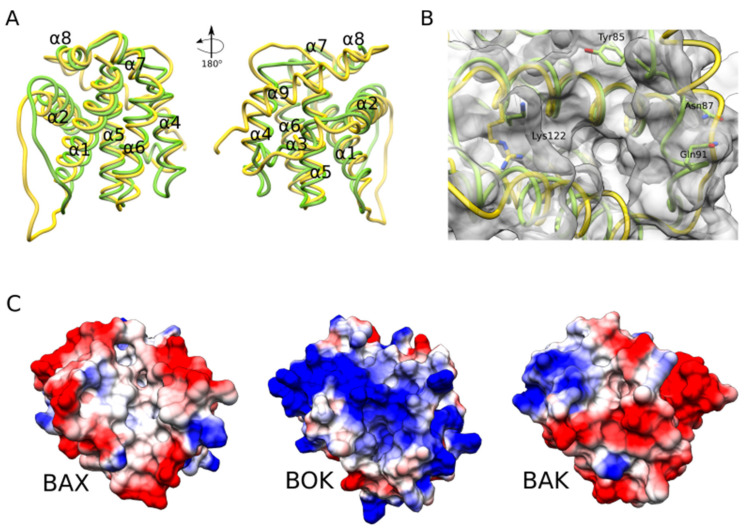
Insights from BOK NMR structure. (**A**) Superposition of the structures of BOK (green) and BAX (yellow) showing the similar fold of the two proteins. The helix numbers are indicated with the helix α9 resolved only in BAX and occupies the hydrophobic groove. (**B**) Zoomed view on the hydrophobic groove from A with key residues displayed as sticks. The helix α3 of BOK is distorted and drifts more into the groove compared to BAX. Tyr85 residue is protruding inside the groove, which hinders the binding of the BH3 ligand. Lys122 is special to BOK (Arginine in all other family members) and forms a weaker interaction with the conserved aspartate in the BH3 ligand. (**C**) Electrostatic surface representation (red (−5 kT/e) to blue (+5 kT/e)) of the hydrophobic groove side of BAX, BOK, and BAK. As shown, BOK appears to be more electropositive compared to the other two proteins.

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
