# Peer review of "The Mysteries around the BCL-2 Family Member BOK"

_biomolecules, 2020, doi:10.3390/biom10121638_

Round 1

Reviewer 1 Report

The review presents a nice overview of BOK and the detailed reference to the crystal structure of BOK in relation to its function and mechanism is interesting.

Comments to address 

1) The introduction lacks a clear description of the BOK dependent non-canonical pathways important for MOM permeabilization. Expanding this section may provide a more appropriate introduction to the role of BOK in apoptosis. Additionally, fig 1B does not add to the review on BOK, perhaps the authors may add the relevance of showing the BAX model of MOMP and how it relates to what is known for BOK in MOMP

2) Section 2.3: Non-apoptotic functions of BOK, seems rather incomplete and lacks references to literature. Perhaps, the authors meant it as an introduction to latter sections and thus may need to either add the remaining sections as sub-sections or expand this section with clear evidence from literature. 

3) The role of BOK as a prognostic marker lacks information on what is known regarding expression and regulation of BOK in various cancers and cancer derived cells. It may be suggested the authors expand this section to provide clear reasoning as to why BOK may be useful as a prognostic marker

Author Response

1) The introduction lacks a clear description of the BOK dependent non-canonical pathways important for MOM permeabilization. Expanding this section may provide a more appropriate introduction to the role of BOK in apoptosis. Additionally, fig 1B does not add to the review on BOK, perhaps the authors may add the relevance of showing the BAX model of MOMP and how it relates to what is known for BOK in MOMP

We thank the reviewer for pointing out these aspects. We have now added a more detailed explanation about the ER-stress dependent non-canonical mode of action to induce MOMP at the introduction section.

Regarding figure 1b, along the text we compiled several evidences supporting that BOK structure and its capacity to induce membrane ruptures share several features with that of BAX and BAK, including sequence homology, the tendency to bind anionic lipids like cardiolipin, the activation induced by the BH3-only protein cBID, its structural reorganization in the membrane and the protein/lipid nature of the pore with tunable size. Thus, we consider it appropriate to speculate that BOK may share this activation steps with BAX-type proteins. We have now clarified this in the text.

2) Section 2.3: Non-apoptotic functions of BOK, seems rather incomplete and lacks references to literature. Perhaps, the authors meant it as an introduction to latter sections and thus may need to either add the remaining sections as sub-sections or expand this section with clear evidence from literature. 

We thank the reviewer for the comment. We have modified the text accordingly and the remaining sections were added as sub-sections

3) The role of BOK as a prognostic marker lacks information on what is known regarding expression and regulation of BOK in various cancers and cancer derived cells. It may be suggested the authors expand this section to provide clear reasoning as to why BOK may be useful as a prognostic marker

The section is now expanded.

Reviewer 2 Report

The review written by Raed Shalaby and colleagues specializes on the Bcl-2 family homolog Bok. The subject is interesting and recent new data in this field enlightened some previously unknown functions and mode of actions of this protein supporting the publication of a review on this topic.

I have some minor points which should be addressed before publication.

1) The majority of the differences between Bax/Bak and Bok comes from its non-mitochondrial localization. The authors are presenting this very briefly (lines 131-133. They should go a little deeper into the mechanism and explain why. One of the main hypothesis is based on differences in the C-terminal TM motif between Bax, Bak and Bok (N Echeverry, 2013, CDD). In this regard the figure 2 is suboptimal since TM motifs are absent. I suggest to align multiple full length Bok sequences using ClustalW or Muscle and compare them to Bax and Bak.

2) In the Figure 1 the authors present the mt-DNA release but this should be also discussed in the corresponding figure legend.

3) The sentence “In light of this model, BOK could be the missing piece linking ER stress to intrinsic apoptosis.” lines 153-154 is misleading. Indeed several other studies have demonstrated a link between ER stress and MOMP mainly through the transcriptional activation of BH3 proteins such as Bim or Bik. Thus it would be more appropriate to speak about a new pathway linking ER stress and MOMP.

4)Since Bok is IP3R binding protein and controls the mitochondrial morphology it worths nothing to be more speculative about a possible localization/role of Bok in the MAMs. Indeed, IP3R calcium channels especially the type 1 are enriched in MAMs.

5) A paragraph about the regulation of BOK expression at transcriptional and translational level could be a plus.

Author Response

1) The majority of the differences between Bax/Bak and Bok comes from its non-mitochondrial localization. The authors are presenting this very briefly (lines 131-133. They should go a little deeper into the mechanism and explain why. One of the main hypothesis is based on differences in the C-terminal TM motif between Bax, Bak and Bok (N Echeverry, 2013, CDD). In this regard the figure 2 is suboptimal since TM motifs are absent. I suggest to align multiple full length Bok sequences using ClustalW or Muscle and compare them to Bax and Bak.

We thank the reviewer for this comment. The sequence alignment of the C-terminal TM motifs is now included. The difference between the TMD of BOK and BAK-BAX is now explained in the lines 116-125 too.

2) In the Figure 1 the authors present the mt-DNA release but this should be also discussed in the corresponding figure legend.

We thank the reviewer for pointing this out. We have now included this aspect at the figure legend.

3) The sentence “In light of this model, BOK could be the missing piece linking ER stress to intrinsic apoptosis.” lines 153-154 is misleading. Indeed several other studies have demonstrated a link between ER stress and MOMP mainly through the transcriptional activation of BH3 proteins such as Bim or Bik. Thus it would be more appropriate to speak about a new pathway linking ER stress and MOMP.

We agree with the reviewer and we have now modified the text accordingly. In this regard, one may speculate that the effect of BOK is further regulated by the interaction with BH3 only BIM or BIK.

4)Since Bok is IP3R binding protein and controls the mitochondrial morphology it worths nothing to be more speculative about a possible localization/role of Bok in the MAMs. Indeed, IP3R calcium channels especially the type 1 are enriched in MAMs.

We thank the reviewer for this comment. We agree with the reviewer that Bok is an IP3R binding protein and that one of the receptor variants is mainly (not exclusively) located at the MAMs. On the other hand, despite Bok has a proved role in mitochondrial morphology the mechanism is still not completely understood. Altogether, we considered more cautious to suggest, rather than to affirm, that Bok is located at the MAMs or that it modulates these structures.

5) A paragraph about the regulation of BOK expression at transcriptional and translational level could be a plus.

We thank the reviewer for this comment. These are now covered by the lines 128-135 of the manuscript.